# Morphology and Phylogeny Reveal *Vamsapriyaceae* fam. nov. (*Xylariales*, *Sordariomycetes*) with Two Novel *Vamsapriya* Species

**DOI:** 10.3390/jof7110891

**Published:** 2021-10-21

**Authors:** Ya-Ru Sun, Ning-Guo Liu, Milan C. Samarakoon, Ruvishika S. Jayawardena, Kevin D. Hyde, Yong Wang

**Affiliations:** 1Department of Plant Pathology, College of Agriculture, Guizhou University, Guiyang 550025, China; yarusun5@gmail.com; 2Center of Excellence in Fungal Research, Mae Fah Luang University, Chiang Rai 57100, Thailand; liuningguo11@gmail.com (N.-G.L.); samare.ag.rjt@gmail.com (M.C.S.); ruvi.jaya@yahoo.com (R.S.J.); kdhyde3@gmail.com (K.D.H.); 3School of Science, Mae Fah Luang University, Chiang Rai 57100, Thailand; 4School of Life Science and Technology, Center for Informational Biology, University of Electronic Science and Technology of China, Chengdu 611731, China; 5Innovative Institute of Plant Health, Zhongkai University of Agriculture and Engineering, Haizhu District, Guangzhou 510000, China

**Keywords:** three new taxa, *Ascomycota* genera *incertae sedis*, multi-gene phylogeny, new family, taxonomy

## Abstract

Phylogenetic analyses of combined *LSU*, rpb2, tub2 and *ITS* sequence data of representative *Xylariales* taxa indicated that *Diabolocovidia*, *Didymobotryum* and *Vamsapriya* cluster together and form a distinct clade in *Xylariales*. Morphological comparison also shows their distinctiveness from other families of *Xylariales*. Therefore, we introduce it as a novel family, *Vamsapriyaceae*. Based on morphological characteristics, *Podosporium* and *Tretophragmia*, which were previously classified in *Ascomycota* genera *incertae sedis*, are now included in the *Vamsapriyaceae*. In addition, three *Vamsapriya* species, *V. chiangmaiensis* sp. nov, *V. uniseptata* sp. nov, and *V. indica* are described and illustrated in this paper.

## 1. Introduction

*Xylariales* is a large order with both conspicuous and inconspicuous fruiting bodies, and unitunicate, perithecial ascomycetes [1,2]. Many species of *Xylariales* are saprobes and endophytes [3,4]. Some *Xylariales* species can produce secondary metabolites which are especially important for the pharmaceutical chemical industry [3,5,6].

*Xylariales* was established by Nannfeldt [7] to accommodate the type family *Xylariaceae*, along with *Diatrypaceae*, *Hypocreaceae*, *Hyponectriaceae*, *Lasiosphaeriaceae* and *Polystigmataceae*. The previous classification of *Xylariales* was mainly based on morphology [8,9,10,11,12,13]. With the development of molecular technology, the classification basis of *Xylariales* was gradually diversified [2,14,15,16]. Smith et al. [2] performed the first multigene analysis to find the familial relationships within *Xylariales* and treated the order with seven families. Lumbsch and Huhndorf [17] listed six families in *Xylariales*, while Senanayake et al. [18] revised *Xylariales* and accepted 11 families. Hyde et al. [19] redefined the families of *Sordariomycetes* and accepted 15 families in *Xylariales* based on morphology and multigene analysis, viz. *Barrmaeliaceae*, *Cainiaceae*, *Clypeosphaeriaceae*, *Coniocessiaceae*, *Diatrypaceae*, *Graphostromataceae*, *Hansfordiaceae*, *Hypoxylaceae*, *Induratiaceae*, *Lopadostomataceae*, *Microdochiaceae*, *Polystigmataceae*, *Requienellaceae*, *Xylariaceae* and *Zygosporiaceae*. Hyde et al. [20] introduced *Fasciatisporaceae* to accommodate *Fasciatispora* in *Xylariales*. However, the taxonomic position of many taxa in *Xylariales* are still uncertain, and they are treated as genera *incertae sedis* [20,21]. This may probably be due to monospecific genera with either sexual or asexual morph, with no additional collections and lack of molecular data, and sometimes due to the polyphyletic nature of some genera (such as *Anthostomella* and *Xylaria*) [22,23,24,25].

*Vamsapriya* was introduced by Gawas and Bhat [26] based on abundant asexual morphs of the genus, which is characterized by erect, cylindrical, dark brown, synnematous conidiophores, monotretic, clavate to cylindrical conidiogenous cells, and cylindrical or broadly fusiform or obclavate, brown to dark brown conidia [26,27,28,29,30,31,32,33]. The first sexual morph of *Vamsapriya* was described by Dai et al. [28], which has solitary, immersed ascomata visible as black dots, 8spored, unitunicate asci, and hyaline, fusiform apiospores. They linked the sexual morph of *V. bambusicola* (MFLUCC 11-0637) to the asexual morph of *V. bambusicola* (MFLUCC 11-0477) using *ITS* phylogenies [27,28]. The phylogenetic placement of *Vamsapriya* has been confusing. Dai et al. [27,28] and Jiang et al. [31] accepted *Vamsapriya* into the *Xylariaceae*. However, phylogenetic analyses using broader taxon sampling indicated that *Vamsapriya* was distant from *Xylariaceae* [19,34].

This study aims to resolve the phylogenetic position of *Vamsapriya*. Three *Vamsapriya* collections (*V. chiangmaiensis* sp. nov, *V. uniseptata* sp. nov, and *V. indica*) on bamboo from China and Thailand are described and illustrated herein. *Vamsapriya*, along with *Diabolocovidia* and *Didymobotryum*, formed a distinct monophyletic clade in the combined *LSU*, rpb2, tub2 and *ITS* phylogenetic analyses. A new family, *Vamsapriyaceae*, is thus established. *Podosporium* and *Tretophragmia* are also accepted in *Vamsapriyaceae* based on their morphology of hyphomycetous asexual morph.

## 2. Materials and Methods

### 2.1. Collection, Examination, Isolation and Conservation

Fresh specimens were collected from bamboo in terrestrial habitats in China and Thailand between August 2019 and September 2020. Sample collections and observations were followed by the method described in Senanayake et al. [35]. The samples were stored in envelopes and taken to the laboratory for examination. Morphological observations were done using a stereo microscope (LEICA M125 C, Wetzlar, Germany). The fungal structures were captured using a Nikon ECLIPSE Ni compound microscope (Nikon, Tokyo, Japan) fitted with a NikonDS-Ri2 digital camera (Nikon, Tokyo, Japan). The Tarosoft (R) Image Frame Work software was used to take the measurements. Adobe Photoshop CS6 software (Adobe Systems, San Jose, CA, USA) was used to do photo-plates.

Single spore isolation was carried out to obtain pure cultures following the method described in Senanayake et al. [35]. Germinated spores were transferred to pure potato dextrose agar (PDA) and cultivated under normal light at 26 °C for four weeks. Herbarium specimens were deposited in the Fungarium of Mae Fah Luang University (MFLU), Chiang Rai, Thailand, and the herbarium of the Guizhou Academy of Agriculture Sciences (GZAAS), Guiyang, China. Pure cultures were deposited in the Mae Fah Luang University Culture Collection (MFLUCC) and the Guizhou Culture Collection (GZCC). FacesofFungi (FoF) and Index Fungorum numbers were obtained as described in Jayasiri et al. [36] and Index Fungorum [37].

### 2.2. DNA Extraction, PCR Amplification and Sequencing

Genomic DNA was extracted from fresh fungal mycelia using the Genomic DNA Extraction Kit (GD2416 BIOMIGA, San Diego, CA, USA). Polymerase chain reactions (PCR) were carried out using a BIO-RAD T100 Thermal Cycler in a 20 μL reaction volume which contained 10 μL 2x PCR Master Mix, 7 μL ddH_2_O, 1 μL of each primer, and 1 μL template DNA. The PCR thermal cycle program and primers are given in Table 1. The PCR products were sent for sequencing to SinoGenoMax, Beijing, China.

### 2.3. Phylogenetic Analyses

The sequences used in this study (Table 2) were downloaded from GenBank according to the results of blast searches and previous studies [27,28,29,30,31,32,33]. Alignments for each locus were carried out in MAFFT v7.212 [40]. AliView [41] was used for checking the alignments and changing the format. Terminal ends and ambiguous regions of the alignment were deleted manually. Four single gene alignments were combined using the Sequence Matrix [42].

Single gene analyses were done to compare the topologies and clade stabilities, respectively. Single and combined phylogenies were subjected to Bayesian posterior probability (BYPP), maximum likelihood (ML) and maximum parsimony (MP) analyses. The BYPP analysis was performed in MrBayes v. 3.2.6 [43]. MrModeltest v. 2.3 [44] was used to estimate the best model. GTR+I+G model was chosen for *LSU* and rpb2; SYM+I+G (*Xylariales* analysis) and GTR+G (*Vamsapriya* analysis) models were chosen for *ITS*; HKY+I+G model was chosen for tub2. Six chains were run and trees were sampled every 1000th generation, the temperature value of the heated chain was set at 0.15. The first 25% sampled trees were discarded as “burn-in”, and the remaining trees were used for calculating BYPP in the majority rule consensus tree. The ML analyses were carried out using IQ-TREE [45] on the IQ-TREE web server (http://iqtree.cibiv.univie.ac.at, 6 September 2021) under partitioned models. The best-fit substitution models were determined by W-IQ-TREE [45]: TIM3e+I+G4 for *LSU*; TIM3+F+I+G4 for rpb2; TIM2+F+I+G4 for tub2; SYM+I+G4 for *ITS*. Ultrafast bootstrap analysis was implemented with 1,000 replicates. The MP analyses were carried out with a heuristic search in PAUP v. 4.0 b10 [46]. Bootstrap analysis was used to estimate clade stability, including 1000 replicates, each with 10 replicates of random stepwise addition of taxa [47].

Phylogenetic trees were viewed using FigTree v1.4.4 [48] and modified in Adobe Illustrator CS6 software (Adobe Systems, USA). The sequences generated from our collections were deposited in GenBank.

## 3. Results

### 3.1. Phylogenetic Analyses

In *Xylariales* phylogenetic analyses, the final combined dataset of *Xylariales* consists of 84 strains representing fifteen families along with the outgroup *Amphisphaeria sorbi* (MFLUCC 13-0721) and *A. thailandica* (MFLU 18-0794) in *Amphisphaeriales*. The aligned sequence matrix comprises *LSU* (1–829), rpb2 (830–1875), tub2 (1876–3579) and *ITS* (3580–4305), sequence data for a total of 4305 characters, including coded alignment gaps. Among them, 1894 characters were constant, 374 variable characters were parsimony-uninformative and 2037 characters were parsimony informative. The matrix had 2693 distinct alignment patterns. The BYPP, ML, and MP analyses based on combined sequence data provided similar tree topology. For BYPP, the standard deviation of split frequencies was reached at 0.0099 after 2,980,000 generations. The most likely tree (−ln = 66,531.894) is presented (Figure 1). The MP analysis resulted in two trees with TL = 15,668, CI = 0.302, RI = 0.524, RC = 0.158, HI = 0.698.

The single locus trees (Appendix A) and the multi-locus (*LSU*, rpb2, tub2 and *ITS*) tree (Figure 1) showed similar tree topology. In multigene analyses, *Vamsapriya* species clustered with *Diabolocovidia claustri* and *Didymobotryum rigidum*, and they formed an internal distinct clade with maximum support (ML-bs = 100%, MP-bs = 100%, BYPP = 1.00). *Xylariaceae*, *Induratiaceae* and *Clypeosphaeriaceae* clustered together, which is a sister to *Vamsapriyaceae* without significant support. Moreover, *V. chiangmaiensis* (MFLUCC 21-0065) formed a sister clade to *V. yunnana;* however, the support for this relationship in Figure 1 is extremely poor and does not exist in Figure 2, and *V. uniseptata* (GZCC 21-0892) is sister to *V. indica*. Our isolate MFLUCC 21-0066 grouped in *V. indica* clade with MFLUCC 12-0544 and DLUCC:2062, indicating they are phylogenetically the same species. Two *Anthostomella* (*Xylariaceae*) species, *A. formosa* (MFLUCC 14-0170) and *A. obesa* (MFLUCC 14-0171) formed a distinct clade and is sister to *Cainiaceae*.

The *ITS* based on *Vamsapriya* analyses contained 12 taxa and rooted with *Diabolocovidia claustra* (CBS 146630) and *Didymobotryum rigidum* (JCM-8837). The manually adjusted *ITS* alignment contained 563 characters. The best scoring RAxML tree with a final likelihood value of −1737.963458 is presented (Figure 2). Maximum parsimony analysis comprised 563 characters, of which 446 were constant, 54 were parsimony-informative, and 63 were parsimony-uninformative; the tree length is 184, CI = 0.739, RI = 0.597, RC = 0.441, HI = 0.261. The results showed our strain MFLUCC 21-0066 clustered together with *V. indica* (MFLUCC 12-0544 and DLUCC:2062) with good supports (ML-bs = 94%, MP-bs = 89%, BYPP = 1.00). *Vamsapriya chiangmaiensis* (MFLUCC 21-0065) formed a distinct clade, and *Vamsapriya uniseptata* (GZCC 21-0892) grouped with three *V. indica* (ML-bs = 79%, MP-bs = 61%).

### 3.2. Taxonomy

*Vamsapriyaceae* Y.R. Sun, Yong Wang bis & K.D. Hyde, fam. nov.

Index Fungorum number: IF558620; Facesoffungi number: FoF09926

Etymology: Name reflects the type genus

Type genus: *Vamsapriya* Gawas & Bhat

*Saprobic* on dead wood. Sexual morph: *Ascomata* solitary, scattered, immersed, subglobose, black, ostiolate. *Peridium* thin-walled, brown. *Paraphyses* hyaline, septate. *Asci* 8-spored, unitunicate, cylindrical, short pedicellate, with a J+ apical ring. *Ascospores* apiosporous, fusiform to broad fusiform, hyaline. Asexual morph: Hyphomycetous. *Colonies* on natural substrate effuse, black, velvety. *Mycelium* immersed, septate, branched. *Synnemata* present or absent; when present (*Didymobotryum*, *Podosporium*, *Tretophragmia*, *Vamsapriya*), *synnemata* erect, rigid, dark brown, composed of compact parallel conidiophores. *Conidiophores* erect, straight or curved, cylindrical, dark brown, septate. *Conidiogenous cells* mono- or polytretic, integrated, terminal, clavate to cylindrical, brown. *Conidia* catenate or solitary, acrogenous, simple, pigmented, multi-shaped, septate; when absent (*Diabolocovidia*, adapted from Crous et al. [49]), *conidiophores* micronematous, flexuous, mostly reduced to a terminal conidiogenous cell. *Conidiogenous cells* monoblastic, subcylindrical to clavate, pale brown, smooth. *Conidia* catenate, acrogenous, brown, ellipsoid to obovoid, thin-walled, aseptate.

Notes: A new family, *Vamsapriyaceae*, is introduced to accommodate *Diabolocovidia*, *Didymobotryum*, *Podosporium*, *Tretophragmia*, and *Vamsapriya*. Their phylogenetic position, which is distinct from other families, supports the establishment of the new family within *Xylariales*. Although the phylogeny of *Podosporium* and *Tretophragmia* could not be inferred due to the lack of molecular data, their morphological characters resemble *Didymobotryum* and *Vamsapriya* in having brown to dark, simple, straight synnemata, monotretic conidiogenous cells and solitary, obclavate, multi-septate, dark brown conidia [50,51,52,53]. We thus temporarily accept *Podosporium* and *Tretophragmia* in *Vamsapriyaceae* based on morphology. Sequence data are needed to resolve their phylogenetic affinities.

*Vamsapriya* Gawas & Bhat, Mycotaxon 94: 150 (2006) [2005]

Index Fungorum number: IF29041; Facesoffungi number: FoF00372

Type species: *Vamsapriya indica* Gawas & Bhat, Mycotaxon 94: 150 (2006) [2005]

*Saprobic* on dead wood. Sexual morph: *Ascomata* solitary, scattered, immersed, subglobose, black, ostiolate. *Peridium* thin-walled, brown. *Paraphyses* hyaline, septate. *Asci* 8-spored, unitunicate, cylindrical, straight, short pedicellate, with a J+ apical ring. *Ascospores* uniseriate or overlapping uniseriate, fusiform to broad fusiform, apiosporous, hyaline, pointed at both ends, surrounded by a mucilaginous sheath. Asexual morph: Hyphomycetous. *Colonies* on natural substrate effuse, black, velvety. *Mycelium* immersed, septate, branched. *Conidiophores* macronematous, synnematous, erect, straight or curved, dark brown, cylindrical, septate. *Synnemata* erect, rigid, dark brown, composed of compact parallel conidiophores. *Conidiogenous cells* monotretic, integrated, terminal, clavate to cylindrical. *Conidia* catenate or solitary, acrogenous, cylindrical, oblong, fusiform or obclavate, brown to dark brown, septate, verruculose.

Notes: *Vamsapriya* species are reported from tropical and subtropical regions, and most species are found in terrestrial as saprobes [26,27,28,29,30,31]. Nine species are accepted in the *Vamsapriya*, of which six have molecular data. *Vamsapriya* is the only holomorphic genus in *Vamsapriyaceae*, and *V. bambusicola* is the only species with a sexual-asexual connection in *Vamsapriya*. Bamboo seems to be the host preference for *Vamsapriya* species [26,27,28,29,30,31,32,33].

*Vamsapriya indica* Gawas & Bhat, Mycotaxon 94: 150 (2006) [2005]

Index Fungorum number: IF550801; Facesoffungi number: FoF00374, Figure 3

*Saprobic* on dead bamboo culms. Sexual morph: Undetermined. Asexual morph: Hyphomycetous. *Colonies* effuse, dark brown, hairy. *Conidiophores* macronematous, synnematous, single, erect, cylindrical, straight or slightly flexuous, dark brown, smooth-walled. *Synnemata* erect, straight or slightly flexuous, dark brown, rigid, with cylindrical to clavate apical fertile part, composed of compactly arranged conidiophores, 1300–1900 um long, 80–150 μm wide at the base, 30–40 μm wide in the middle, 60–140 μm wide at the apical fertile region, with basal portion immersed. *Conidiogenous cells* monotretic, integrated, terminal, brown, cylindrical to clavate, apically rounded, smooth-walled, 4.5–8.5 × 3–4.5 μm ($\bar{x}$ = 6.5 × 4 μm, *n* = 30). *Conidia* catenate, acrogenous, cylindrical, rounded at the apex, taper and subtruncate at the base, olivaceous brown to brown, 2–8-septate, slightly constricted at the septa, smooth, 20–48 × 4.5–6.5 μm ($\bar{x}$ = 32 × 5.5 μm, *n* = 20).

Cultural characters: Conidia germinated on PDA within 12 h, germ tubes produced from both ends. Colonies reached 20 mm diam. within four weeks at 26 °C, cottony, flat, circular, edge entire, white from above, white to yellow from the below.

Material examined: Thailand, Chiang Mai Province, Mae Taeng District, Pa Pae, Mushroom Research Center, on bamboo culms, 10 September 2020, H.W. Shen, M38 (MFLU 21-0088; living culture, MFLUCC 21-0066).

Notes: *Vamsapriya indica* is the type species of *Vamsapriya* [26]. Dai et al. [27] recollected *V. indica* from Thailand and provided the culture characters and sequences data. Bao et al. [32] reported it from a bamboo plant in a freshwater habitat in China. Including our collection, all of these four isolates are recorded from bamboo. Morphological comparison is shown in Table 3. Our collection has longer synnemata than those of the three isolates.

*Vamsapriya chiangmaiensis* Y.R. Sun, Yong Wang bis & K.D. Hyde, sp. nov.

Index Fungorum number: IF558618; Facesoffungi number: FoF09927, Figure 4

Etymology: Name reflects the collected site.

Holotype: MFLU 21-0087

*Saprobic* on dead bamboo culms. **Sexual morph:** *Ascomata* 650–1000 × 650–850 μm, solitary scattered, immersed within the host cortex, visible as black, circular dots, in cross section globose to subglobose. *Ostiole* raised, centric, periphysate ostiolar canal. *Peridium* composed of hyaline inner layer and dark brown to dark outer layer. *Paraphyses* long, hyaline, unbranched, septate, 1.5–4 μm wide ($\bar{x}$ = 2 μm, *n* = 15). *Asci* 8-spored, unitunicate, straight or slightly curved, cylindrical, short pedicellate, with apical ring, 140–190 × 6.5–12 μm ($\bar{x}$ = 160 × 9 μm, *n* = 15). *Ascospores* uniseriate, fusiform, 17–26 × 5.5–8 μm ($\bar{x}$ = 20.5 × 6.5 μm, *n* = 30), constricted apiosporous with a large cell 12.5–22 μm length, guttulate; basal cell 3.5–6.5 μm length, hyaline, smooth-walled, surround a gelatinous mucilaginous sheath. Asexual morph: Undetermined.

Culture characters: Ascospores germinated on PDA within 12 h, germ tubes produced from one end. Colonies reached 45 mm diam. within four weeks at 26 °C, flat, circular, cottony. White from above; brown to dark brown in the center, white to pale brown around from below.

Material examined: Thailand, Chiang Mai Province, Mae Taeng District, Mushroom Research Center, on bamboo culms, 15 July 2020, Y.R. Sun, M35 (MFLU 21-0087, holotype; ex-type living culture, MFLUCC 21-0065).

Notes: *Vamsapriya chiangmaiensis* is the second species that has a sexual morph in *Vamsapriya*. It is similar to *V. bambusicola* in having solitary, subglobose ascomata, 8-spored, unitunicate, cylindrical asci and fusiform hyaline ascospores. It can be distinguished by the longer asci (140–190 μm vs. 115–140 μm). In addition, polymorphic nucleotides from the *ITS* region showed 37 base differences, and the details are given in Table 4. Therefore, we identified *V. chiangmaiensis* as a new species following the suggestions for species delineation [54].

*Vamsapriya uniseptata* N.G. Liu & K.D. Hyde , sp. nov.

Index Fungorum number: IF558619; Facesoffungi number: FoF09928, Figure 5.

Etymology: Name reflects the 1-septate conidia.

Holotype: GZAAS 21-0378

*Saprobic* on submerged decaying wood in terrestrial habitat. *Colonies* on natural substrate effuse, black, velvety. Asexual morph: Hyphomycetous. *Mycelium* mostly immersed, composed of septate, branched, hyaline to brown hyphae. *Conidiophores* macronematous, synnematous, erect, straight or broadly curved, dark brown, cylindrical, septate. *Synnemata* erect, rigid, dark brown, composed of compact parallel conidiophores, up to 1300 µm long, 30–50 µm wide in the middle. *Conidiogenous cells* monotretic, integrated, terminal, clavate, brown to dark brown. *Conidia* catenate, acrogenous, olivaceous brown, smooth, oblong, rounded at the apex, taper and subtruncate at the base, 1-septate at the middle, septa thickened and darkened, slightly constricted at the septa, with a large globule in each cell, 14–19 × 3.5–4.5 μm ($\bar{x}$ = 16.5 × 5 µm, *n* = 30). Sexual morph: Unknown.

Cultural characters: Conidia germinated on PDA within 12 h and germ tubes produced from both ends. Colonies reached 30 mm within four weeks at 26 °C, flat, circular, cottony, white from above, from below brown to dark brown in the center, white to pale brown around.

Material examined: China, Guizhou Province, Xingyi City, Qingshuihe Town, 8 August 2019, N.G. Liu, Q1 (GZAAS 21-0378, holotype; ex-type living culture, GZCC 21-0892).

Notes: *Vamsapriya uniseptata* is distinguishable by having smaller, 1-septate conidia, while other *Vamsapriya* species have elongated phragmoconidia. In the BLASTn search, the closest match of the *ITS* sequence of *V. uniseptata* is *V. khunkonensis* (MFLUCC 13-0497, MFLUCC 11-0475 (93.4%)), followed by *V. indica* (MFLUCC 12-0544 (91.7%)). The *LSU* sequence of *V. uniseptata* is *V. indica* (DLUCC:2062 (99.8%)) and *V. khunkonensis* (MFLUCC 11-0475 (99.7%)). *Vamsapriya uniseptata* can be distinguished from *V. khunkonensis* in the multigene phylogenetic analyses. The *ITS* region of *V. indica* (MFLUCC 13-0497) differs by 23 base pairs (527 bp without gaps). Based on distinct morphology and phylogeny, *V. uniseptata* is introduced as a novel taxon.

### 3.3. Other Accepted Genera in Vamsapriyaceae

*Diabolocovidia* Crous, Persoonia 44: 331 (2020)

Index Fungorum number: IF835401; Facesoffungi number: FoF09929.

*Parasitic* on leaves in terrestrial habitats. *Mycelium* composed of septate, branched, hyaline to pale brown hyphae. Asexual morph: Hyphomycetous. *Conidiophores* solitary, erect, flexuous, mostly reduced to a terminal conidiogenous cell. *Conidiogenous cells* monoblastic, terminal, subcylindrical to clavate, pale brown, smooth. *Conidia* catenate, acrogenous, brown, ellipsoid to obovoid, thin-walled, un-septate, verruculose [49]. Sexual morph: Unknown.

Type species: *Diabolocovidia claustri* Crous

Notes: *Diabolocovidia* is a monotypic genus introduced by Crous et al. [49] to accommodate *Diabolocovidia claustri*, which was isolated from leaves of *Serenoa repens* in the U.S.A. *Diabolocovidia claustri* is characterized by mononematous, micronematous conidiophores in *Xylariaceae*. In their phylogenetic analyses, *Diabolocovidia* is basal to *Vamsapriya* [49]. *Diabolocovidia* is the only genus without synnemata in *Vamsapriyaceae*.

*Didymobotryum* Sacc., Syll. fung. (Abellini) 4: 626 (1886)

Index Fungorum number: IF8009; Facesoffungi number: FoF09930.

*Saprobic* on decaying plants materials in terrestrial habitats. *Colonies* on natural substrate effuse, olivaceous to dark brown, velvety. *Mycelium* mostly immersed, composed of septate, branched, thick-walled, subhyaline hyphae. Asexual morph: Hyphomycetous. *Conidiophores* macronematous, synnematous, erect, straight or broadly curved, dark brown, cylindrical, septate. *Synnemata* erect, rigid, dark brown, composed of compact parallel conidiophores. *Conidiogenous cells* monotretic, integrated, integrated or discrete, cylindrical to clavate, olivaceous brown to dark brown. *Conidia* catenate, dry, acrogenous, cylindrical, verrucose, 1-septate, slightly constricted at the septa, olivaceous brown to brown. Sexual morph: Unknown.

Type species: *Didymobotryum rigidum* (Berk. & Broome) Sacc.

Notes: *Didymobotryum* was introduced by Saccardo [55] typified by *D. rigidum*. *Didymobotryum* species have a worldwide distribution [56,57,58,59]. Six species are accepted in the Index Fungorum [37] but only *D. rigidum* has molecular data (*ITS*: LC228650, *LSU*: LC228707).

*Podosporium* Schwein., Trans. Am. phil. Soc., New Series 4(2): 278 (1832) [1834]

Index Fungorum number: IF9487; Facesoffungi number: FoF09931.

*Saprobic* on decaying plants materials in terrestrial habitats. *Colonies* on natural substrate effuse, brown, velvety. *Mycelium* mostly immersed, composed of septate, flexuous branched hyphae. Asexual morph: Hyphomycetous. *Conidiophores* arranged in synnemata, brown, septate, sometimes branched at the apex. *Synnemata* erect, rigid, brown to dark. *Conidiogenous cells* mono- or polytretic, integrated or discrete, subulate or cylindrical, darkly pigmented. *Conidia* solitary, obclavate or bacilliform, multi-septate, brown to dark brown. Sexual morph: Unknown.

Type species: *Podosporium rigidum* Schwein.

Notes: *Podosporium* was introduced by Schweinitz [60] with *P. rigidum* as the type species. Since then, many *Podosporium* species have been discovered worldwide [60,61,62,63]. Most of them are saprobes in terrestrial habitats [60,61,62,63]. There are 67 species listed in the Index Fungorum [37] but no sequence data are available.

*Tretophragmia* Subram. & Natarajan, Proc. Natl. Inst. Sci. India, B, Biol. Sci. 39: 550 (1974) [1973]

Index Fungorum number: IF10265; Facesoffungi number: FoF09932.

*Saprobic* on plants materials in terrestrial habitats. *Colonies* on natural substrate effuse, dark, velvety. Asexual morph: Hyphomycetous. *Conidiophores* macronematous, synnematous, brown, septate, erect, straight or broadly curved. *Synnemata* rigid, brown to dark, simple, erect, straight, consisting of a stalk and a capitate, broadened, fertile head. *Conidiogenous cells* monotretic, subulate or cylindrical, darkly pigmented. *Conidia* solitary, obclavate to fusiform or irregular in shape, straight, curved or bent, multi-septate, dark brown. Sexual morph: Unknown.

Type species: *Tretophragmia nilgirensis* (Subram.) Subram. & Natarajan

Notes: *Tretophragmia* was introduced in 1974. The asexual morph of *Tretophragmia* is similar to *Didymobotryum*, *Podosporium* and *Vamsapriya*, while no sexual morph is reported. Seifert et al. [53] treated *Tretophragmia* as a synonym of *Podosporium*. However, *Tretophragmia* is accepted in the Index Fungorum [37] and the MycoBank [64] as a separate genus. So far, only two species of *Tretophragmia* have been described [37] and no sequence data are available. Thus, based on morphology and until DNA sequences data are available, we regard this as a separate genus.

## 4. Discussion

In this study, three *Vamsapriya* species, *V. chiangmaiensis*, *V. indica* and *V. uniseptata* were collected from bamboo in terrestrial habitats. In our phylogenetic analyses of combined *LSU*, rpb2, tub2 and *ITS* sequence data, *Diabolocovidia*, *Didymobotryum* and *Vamsapriya* formed a distinct clade in *Xylariales*. Morphological comparison also shows their distinctiveness from other families in *Xylariales*. Therefore, we propose *Vamsapriyaceae* as a new family in *Xylariales.* The sexual morph of *Vamsapriya* differs from those of *Xylariaceae* in having hyaline apiospores [28,30]. It is noteworthy that the sexual morph of *Vamsapriya* is similar to *Induratiaceae* in having 8-spored asci with J+ apical ring and hyaline, apiospores, but *Induratia* (*Induratiaceae*) differs in having geniculosporium asexual morphs [34]. *Apioclypea* is morphologically similar to the sexual morph of *Vamsapriya* in having 8-spored, pedunculate, cylindrical asci and biseriate, fusiform, hyaline ascospores with a mucilaginous sheath, but its asexual morph is unknown [19,21].

*Clypeosphaeriaceae* and *Induratiaceae* are two other families that are phylogenetically related to *Vamsapriyaceae*, but they are distinct in morphology. *Apioclypea*, *Aquasphaeria*, *Brunneiapiospora*, *Clypeosphaeria*, *Crassoascus*, and *Palmaria* (*Clypeosphaeriaceae*) lack asexual morph descriptions and *Diabolocovidia*, *Didymobotryum*, *Podosporium* and *Tretophragmia* (*Vamsapriyaceae*) do not have sexual morph descriptions for the comparisons in Table 5 and Table 6.

*Diabolocovidia claustri* was isolated on leaves of *Serenoa repens* by Crous et al. [49]. Although it has a close phylogenetic relationship with *Vamsapriya*, they are quite different in morphology. *Diabolocovidia* has micronematous rather than synnematous conidiophores, blastic rather than tretic conidiogenous cells, and ellipsoid to obovoid, aseptate conidia [49]. The phenomenon that *Diabolocovidia* mixes with synnematous and tretic genera like *Didymobotryum* and *Vamsapriya* reminds us of an example that *Vanakripa* with blastic conidiogenous resides in the phialidic genus *Conioscypha* [65]. These probably indicate the polyphyletic nature of some hyphomycetous groups. However, since *D. claustri* is the only species represented by one isolate in *Diabolocovidia*, we suggest using more collections to confirm its phylogenetic placement in the future.

When introducing *Vamsapriya*, Gawas and Bhat [26] pointed out *Vamsapriya* (conidia catenate, cylindrical to vermiform, phragmosporous) exhibits a combination of morphological characters of *Didymobotryum* (conidia catenate, ellipsoidal-cylindrical, didymosporous) [51,53,54] and *Podosporium* (conidia solitary, obclavate, phragmosporous) [56,61,63]. However, as more species are added to these three genera, such generic concepts based on conidial morphology have been dispelled. For example, *V. uniseptata* resembles species of *Didymobotryum* in having catenate, oblong, and 1-septate conidia, but it clusters with the type species of *Vamsapriya*, *V. indica. Vamsapriya breviconidiophora* and *V. yunnana* resemble *Podosporium* species in having obclavate, solitary, and multi-septate conidia, but they are grouped with *V. aquatica*, which has catenate, cylindrical to obclavate, multi-septate conidia in the phylogenetic tree. Either the authors did not follow the generic concepts strictly when introducing species, or these three genera are probably congeneric. We tend to infer the latter; however, the conclusion requires a detailed re-examination of herbarium specimens and molecular data.

## Figures and Tables

**Figure 1 jof-07-00891-f001:**
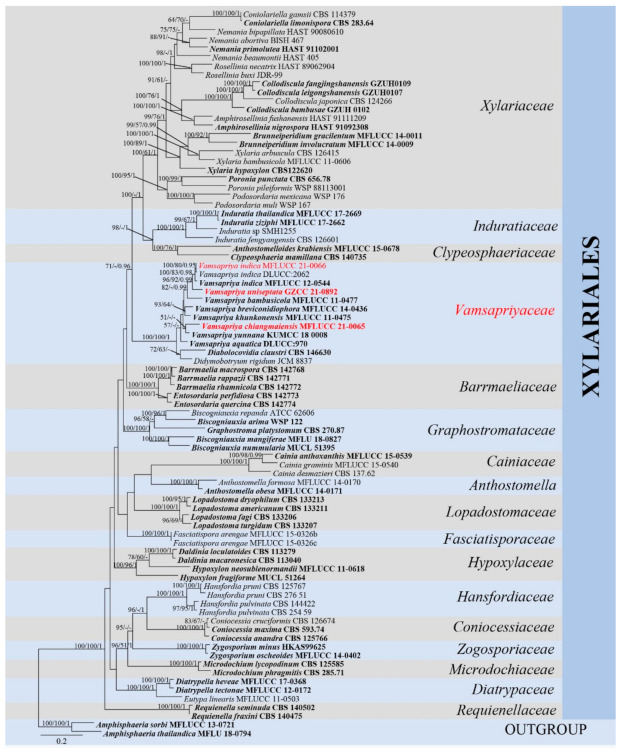
Maximum likelihood (RAxML) tree, based on analysis of a combined dataset of *LSU*, rpb2, tub2 and *ITS* sequence data. The tree is rooted with *Amphisphaeria sorbi* (MFLUCC 13-0721) and *A. thailandica* (MFLU 18-0794). Bootstrap support values for ML and MP greater than 50% and Bayesian posterior probabilities greater than 0.95 are given near nodes, respectively. Ex-type strains are in bold, the new isolates are in red.

**Figure 2 jof-07-00891-f002:**
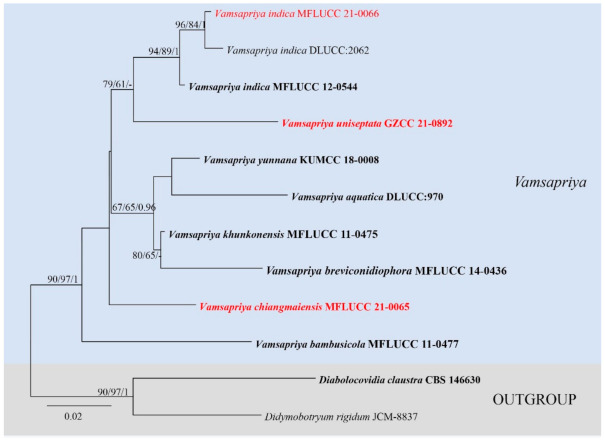
Maximum likelihood (RAxML) tree for *Vamsapriya*, based on *ITS* sequence data. The tree is rooted with *Diabolocovidia claustra* (CBS 146630) and *Didymobotryum rigidum* (JCM-8837). Ex-type strains are in bold, the new isolates are in red.

**Figure 3 jof-07-00891-f003:**
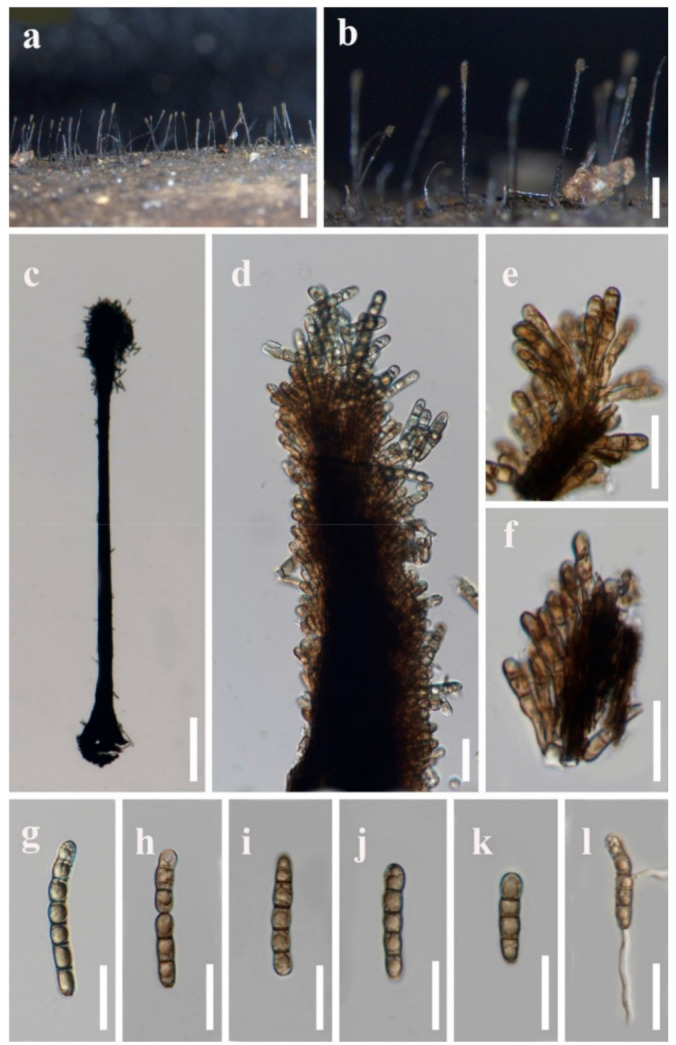
*Vamsapriya indica* (MFLU 21-0088) (**a**,**b**) Colonies on natural substrate. (**c**) Conidiophore and conidia. (**d**–**f**) Conidiogenous cells and developing conidia. (**g**–**k**) Conidia. (**l**) Germinated conidium. Scale bars: **a** = 2000 µm, **b** = 1000 µm, **c** = 200 µm, **d**–**l** = 20 µm.

**Figure 4 jof-07-00891-f004:**
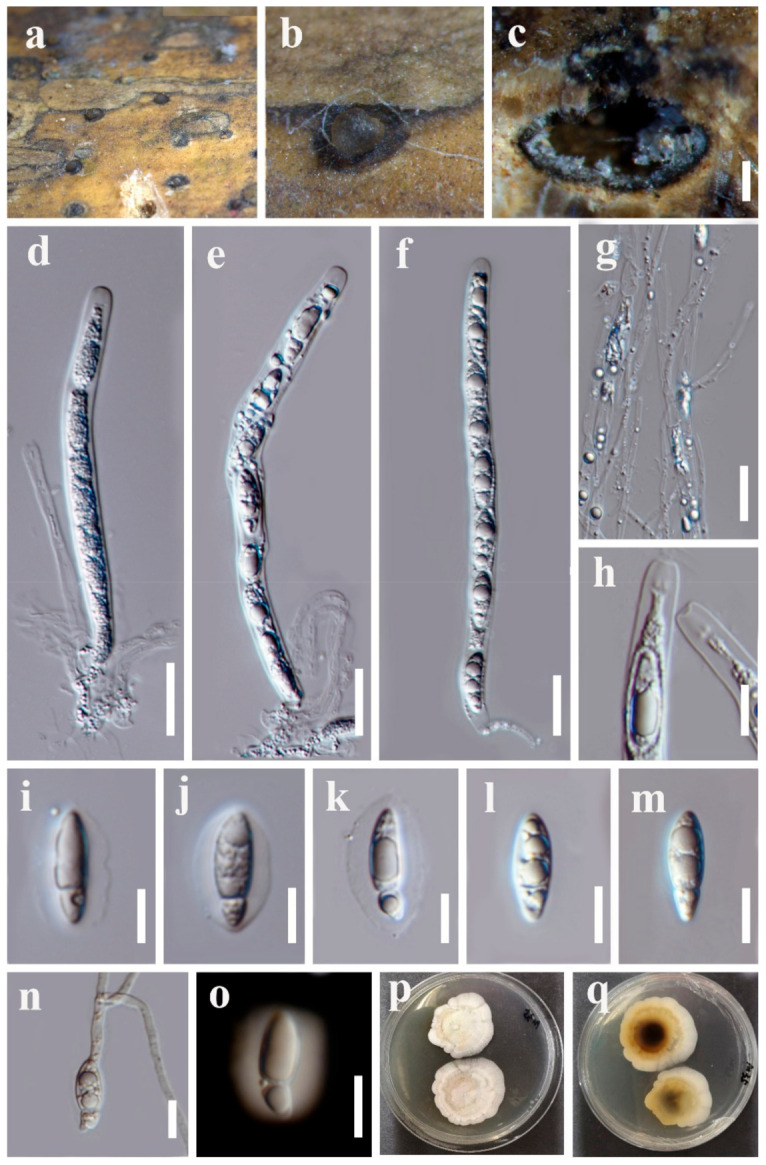
*Vamsapriya chiangmaiensis* (MFLU 21-0087, holotype) (**a**,**b**) Appearance of ascomata on host substrate. (**c**) Vertical section of ascoma. (**d**–**f**) Asci. (**g**) Paraphyses. (**h**) Apical ring of asci. (**i**–**m**) Ascospores. (**n**) Germinated ascospore. (**o**) Ascospore stained in Indian ink. (**p**,**q**) Colonies on PDA. Scale bars: **c** = 200 μm, **d**–**g** = 20 μm, **f**,**h**–**o** = 10 µm.

**Figure 5 jof-07-00891-f005:**
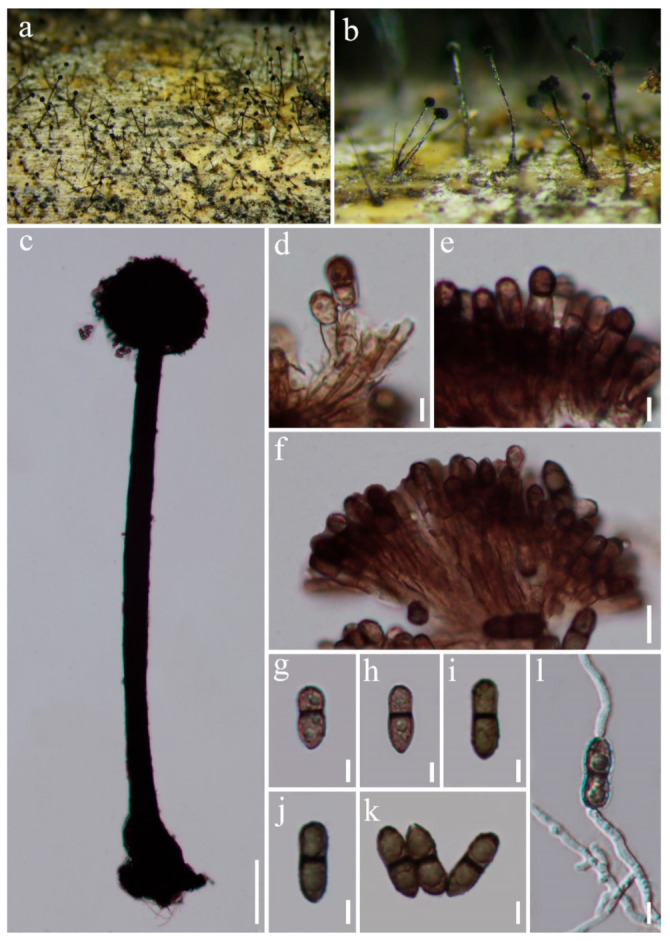
*Vamsapriya uniseptata* (GZAAS:21-0378, holotype) (**a**,**b**) Colonies on natural substrate. (**c**) Conidiophores and conidia. (**d**–**f**) Conidiogenous cells and developing conidia. (**g**–**k**) Conidia. (**l**) Germinated conidium. Scale bars: **c** = 100 µm, **d**,**e**,**g**,**h** = 5 µm, **f** = 10 µm.

**Table 1 jof-07-00891-t001:** Primers and PCR protocol used in this study.

Locus	Primers	PCR Procedure	Reference
*LSU*	LR0R	94 °C 3 min; 35 cycles of 94 °C 30 s, 52 °C 30 s, 72 °C 1 min; 72 °C 8 min; 4 °C on hold	[38,39]
	LR5		
*ITS*	ITS5		
	ITS4		

**Table 2 jof-07-00891-t002:** Taxa names, strain numbers and corresponding sequences used for the molecular phylogenetic analyses.

Taxa	Strain Numbers	*ITS*	*LSU*	rpb2	tub2
*Amphirosellinia fushanensis*	HAST 91111209	GU339496	N/A	GQ848339	GQ495950
** *Amphirosellinia nigrospora* **	**HAST 91092308**	**GU322457**	N/A	**GQ848340**	**GQ495951**
** *Amphisphaeria sorbi* **	**MFLUCC 13-0721**	**NR_153531**	**KP744475**	N/A	N/A
** *Amphisphaeria thailandica* **	**MFLU 18-0794**	**NR_168783**	**NG_068588**	**MK033640**	**MK033639**
*Anthostomella formosa*	MFLUCC 14-0170	KP297403	KP340544	KP340531	KP406614
** *Anthostomella obesa* **	**MFLUCC 14-0171**	**KP297405**	**KP340546**	**KP340533**	**KP406616**
** *Anthostomelloides krabiensis* **	**MFLUCC 15-0678**	**KX305927**	**KX305928**	**KX305929**	N/A
** *Barrmaelia rhamnicola* **	**CBS 142772**	**NR_153497**	N/A	**MF488999**	**MF489018**
** *Barrmaelia rappazii* **	**CBS 142771**	**NR_153496**	N/A	**MF488998**	**MF489017**
** *Barrmaelia macrospora* **	**CBS 142768**	**NR_167684**	N/A	**MF488995**	**MF489014**
** *Biscogniauxia arima* **	**WSP 122**	**NR_167683**	N/A	**GQ304736**	**AY951672**
** *Biscogniauxia mangiferae* **	**MFLU 18-0827**	**MN337232**	**MN336236**	**MN366247**	**MN509782**
** *Biscogniauxia nummularia* **	**MUCL 51395**	**NR_153649**	**NG_066378**	**KY624236**	**KX271241**
*Biscogniauxia repanda*	ATCC 62606	KY610383	KY610428	N/A	KX271242
** *Brunneiperidium gracilentum* **	**MFLUCC 14-0011**	**KP297400**	**KP340542**	**KP340528**	**KP406611**
** *Brunneiperidium involucratum* **	**MFLUCC 14-0009**	**KP297399**	**KP340541**	**KP340527**	**KP406610**
** *Cainia anthoxanthis* **	**MFLUCC 15-0539**	**NR_138407**	**NG_070382**	N/A	N/A
*Cainia graminis*	MFLUCC 15-0540	KR092793	KR092781	N/A	N/A
*Cainia desmazieri*	CBS 137.62	MH858124	MH869702	N/A	N/A
** *Clypeosphaeria mamillana* **	**CBS 140735**	N/A	**NG_067338**	**MF489001**	**MH704637**
** *Collodiscula bambusae* **	**GZUH 0102**	**KP054279**	**KP054280**	**KP276675**	N/A
*Collodiscula japonica*	CBS 124266	JF440974	JF440974	KY624273	KY624316
** *Collodiscula fangjingshanensis* **	**GZUH0109**	**KR002590**	**KR002591**	**KR002592**	**KR002589**
** *Collodiscula leigongshanensis* **	**GZUH0107**	**KP054281**	**KP054282**	**KR002588**	**KR002587**
** *Coniocessia anandra* **	**CBS 125766**	**MH863747**	**MH875215**	N/A	N/A
** *Coniocessia maxima* **	**CBS 593.74**	**NR_137751**	**NG_070051**	N/A	N/A
*Coniocessia cruciformis*	CBS 126674	MH864206	MH875663	N/A	N/A
** *Coniolariella limonispora* **	**CBS 283.64**	**KF719198**	**KF719210**	N/A	N/A
*Coniolariella gamsii*	CBS 114379	GU553325	GU553329	N/A	N/A
** *Daldinia macaronesica* **	**CBS 113040**	**JX658504**	**KY610477**	**KY624294**	**KX271266**
** *Daldinia loculatoides* **	**CBS 113279**	**MH862918**	**MH874491**	**KY624247**	**KX271246**
** *Diabolocovidia claustri* **	**CBS 146630**	**MT373367**	**MT373350**	N/A	N/A
** *Diatrypella heveae* **	**MFLUCC 17-0368**	**NR_154046**	**NG_069531**	N/A	**MG334557**
** *Diatrypella tectonae* **	**MFLUCC 12-0172**	**NR_154029**	**NG_069423**	N/A	**KY421043**
*Didymobotryum rigidum*	JCM 8837	LC228650	LC228707	N/A	N/A
** *Entosordaria quercina* **	**CBS 142774**	**NR_153499**	N/A	**MF489004**	**MF489022**
** *Entosordaria perfidiosa* **	**CBS 142773**	**NR_153498**	N/A	**MF489003**	**MF489021**
*Eutypa linearis*	MFLUCC 11-0503	KU940150	KU863138	N/A	N/A
** *Fasciatispora arengae* **	**MFLUCC 15-0326b**	**MK120301**	**MK120276**	N/A	N/A
** *Fasciatispora arengae* **	**MFLUCC 15-0326c**	**MK120302**	**MK120277**	N/A	N/A
** *Graphostroma platystomum* **	**CBS 270.87**	**JX658535**	N/A	N/A	**HG934108**
*Hansfordia pulvinata*	CBS 254.59	KF893288	MH869394	N/A	N/A
*Hansfordia pulvinata*	CBS 144422	MK442587	MK442527	N/A	N/A
*Hansfordia pruni*	CBS 125767	MH863748	MH875216	N/A	N/A
*Hansfordia pruni*	CBS 276.51	MH856854	MH868374	N/A	N/A
** *Hypoxylon fragiforme* **	**MUCL 51264**	**KC477229**	**NG_066364**	N/A	**KX271282**
** *Hypoxylon neosublenormandii* **	**MFLUCC 11-0618**	**NR_155174**	**NG_066168**	N/A	N/A
*Induratia* sp.	SMH1255	MN250031	AY780069	N/A	AY780119
*Induratia fengyangensis*	CBS 126601	HM034852	HM034858	HM034847	HM034839
** *Induratia ziziphi* **	**MFLUCC 17-2662**	**MK762705**	**MK762712**	**MK791281**	**MK776958**
** *Induratia thailandica* **	**MFLUCC 17-2669**	**MK762707**	**MK762714**	**MK791283**	**MK776960**
** *Lopadostoma dryophilum* **	**CBS 133213**	**NR_132028**	N/A	**KC774526**	**MF489023**
** *Lopadostoma americanum* **	**CBS 133211**	**NR_132027**	N/A	N/A	N/A
** *Lopadostoma fagi* **	**CBS 133206**	**NR_132029**	N/A	**KC774531**	N/A
** *Lopadostoma turgidum* **	**CBS 133207**	**NR_132036**	N/A	**KC774562**	**MF489024**
** *Microdochium lycopodinum* **	**CBS 125585**	**NR_145223**	**KP858952**	**KP859125**	**KP859080**
** *Microdochium phragmitis* **	**CBS 285.71**	**NR_132916**	**NG_058147**	**KP859122**	**KP859077**
*Nemania abortiva*	WSP 71221	GU292816	N/A	GQ844768	GQ470219
*Nemania beaumontii*	HAST 405	GU292819	N/A	GQ844772	GQ470222
*Nemania bipapillata*	HAST 90080610	GU292818	N/A	GQ844771	GQ470221
** *Nemania primolutea* **	**HAST 91102001**	**EF026121**	N/A	**GQ844767**	**EF025607**
*Podosordaria mexicana*	WSP 176	GU324762	N/A	GQ853039	GQ844840
*Podosordaria muli*	WSP 167	GU324761	N/A	GQ853038	GQ844839
*Poronia pileiformis*	WSP 88113001	GU324760		GQ853037	GQ502720
** *Poronia punctata* **	**CBS 656.78**	**KT281904**	**KY610496**	**KY624278**	**KX271281**
** *Requienella fraxini* **	**CBS 140475**	**NR_138415**	N/A	N/A	N/A
** *Requienella seminuda* **	**CBS 140502**	**NR_154630**	**MH878683**	**MK523300**	N/A
*Rosellinia buxi*	JDR-99	GU300070	N/A	GQ844780	GQ470228
*Rosellinia necatrix*	HAST 89062904	EF026117	KF719204	GQ844779	EF025603
** *Vamsapriya aquatica* **	**DLUCC:970**	**MZ420740**	N/A	N/A	N/A
** *Vamsapriya bambusicola* **	**MFLUCC 11-0477**	**KM462835**	**KM462836**	**KM462834**	**KM462833**
** *Vamsapriya breviconidiophora* **	**MFLUCC 14-0436**	**MF621584**	**MF621588**	N/A	N/A
** *Vamsapriya chiangmaiensis* **	**MFLUCC 21-0065**	**MZ613171**	**MZ613168**	N/A	N/A
** *Vamsapriya indica* **	**MFLUCC 12-0544**	**KM462839**	**KM462840**	**KM462841**	**KM462838**
*Vamsapriya indica*	DLUCC: 2062	MZ420747	MZ420762	MZ442699	N/A
*Vamsapriya indica*	MFLUCC 21-0066	MZ613172	MZ613169	OK560921	N/A
** *Vamsapriya khunkonensis* **	**MFLUCC 13-0497**	**KM462830**	**KM462831**	**KM462829**	**KM462828**
** *Vamsapriya uniseptata* **	**GZCC 21-0892**	**MZ613173**	**MZ613170**	N/A	N/A
** *Vamsapriya yunnana* **	**KUMCC 18-0008**	**MG833874**	**MG833873**	**MG833875**	N/A
*Xylaria arbuscula*	CBS 126415	KY610394	KY610463	KY624287	KX271257
*Xylaria bambusicola*	MFLUCC 11-0606	KU940160	KU863148	KU940183	N/A
** *Xylaria hypoxylon* **	**CBS122620**	**AM993141**	**KM186301**	**KM186302**	**KX271279**
** *Zygosporium oscheoides* **	**MFLUCC 14-0402**	**MF621585**	**MF621589**	N/A	N/A
** *Zygosporium minus* **	**HKAS99625**	**MF621586**	**MF621590**	N/A	N/A

Abbreviations: ATCC: American Type Culture Collection, Virginia, USA; CBS: Centraalbureau voor Schimmelcultures, Utrecht, Netherlands; CPC: Culture collection of Pedro Crous, housed at CBS; GZCC: Guizhou Culture Collection, Guiyang, China; GZUH: The herbarium of Guizhou University, Guiyang, China; HAST: Herbarium, Research Center for Biodiversity, Academia Sinica, Taipei, China; HKAS: The Herbarium of Cryptogams, Kunming Institute of Botany Academia Sinica, Kunming, China; JDR: Herbarium of Jack D. Rogers; KUMCC: The Kunming Institute of Botany Culture Collection, Kunming, China; MFLU: The Fungarium of Mae Fah Luang University, Chiang Rai, Thailand; MFLUCC: Mae Fah Luang University Culture Collection, Chiang Rai, Thailand; MUCL: Mycothèque de l’Université Catholique de Louvian, Belgium; WSP: Washington State University, U.S.A. The newly generated sequences are indicated in red. Ex-type strains are in bold.

**Table 3 jof-07-00891-t003:** Synopsis of characters of *Vamsapriya indica* collections.

Taxon	Host	Habitat/Location	Synnemata	Conidiogenous Cells	Conidia
*V. indica* (M 393674)	Bamboo	Terrestrial/India	700–1100 μm long, 60–160 μm wide at the base, 30–60 μm wide in the middle, 30–80 μm wide at the apical fertile region	Monotretic, clavate, dark brown, 4–9 × 2.5–4.5 μm	Catenate, acrogenous, brown, cylindrical, vermiform, 10–80 × 4–6 μm, 2–12-septate
*V. indica* (MFLU 13-0370)	Bamboo	Terrestrial/Thailand	700–1100 μm long, 60–160 μm wide at the base, 30–60 μm wide in the middle, 30–80 μm wide at the apical fertile region	Monotretic, ellipsoidal, brown to dark brown, 4–9 × 2.5–4.5 μm ($\bar{x}$ = 6.5 × 3.7 μm, *n* = 20)	Catenate, cylindrical, pale brown to dark brown 35–290 × 4–6.5 μm ($\bar{x}$ = 66.6 × 5.6 μm, *n* = 20), 1–3-septate when young, more than 20–septate at maturity
*V. indica* (HKAS 115803)	Bamboo	Freshwater/China	1145–1475 μm long, 105–235 μm wide at the base, 50–80 μm wide in the middle, 70–155 μm wide at the apical fertile region	Monotretic, clavate, dark brown 5–9 × 3–5 μm ($\bar{x}$ = 20 × 5 μm, *n* = 30)	Catenate, brown to dark brown, cylindrical to obclavate, 15–30 × 4–6.5 μm ($\bar{x}$ = 20 × 5 μm, *n* = 30), 1–4-septate
*V. indica* (MFLU 21-0088)	Bamboo	Terrestrial/Thailand	1300–1900 μm long, 80–150 μm wide at the base, 30–40 μm wide in the middle, 60–140 μm wide at the apical fertile region	Cylindrical to clavate, brown, 4.5–8.5 × 3–4.5 μm ($\bar{x}$ = 6.5 × 4 μm, *n* = 30)	Catenate, olivaceous brown to brown, cylindrical, 20–48 × 4.5–6.5 μm ($\bar{x}$ = 32 × 5.5 μm, *n* = 20), 2–8-septate

**Table 4 jof-07-00891-t004:** Nucleotide differences in the *ITS* regions of *V. bambusicola* and *V. chiangmaiensis*. Numbers are in reference to the nucleotide position of DNA sequences (*V. bambusicola*) submitted in GenBank.

Species	*ITS*																		
	42	52	73	74	82	106	127	167	171	173	194	196	203	206	207	208	209	213	214
*V. bambusicola* (MFLUCC 11-0477)	C	T	C	G	T	T	A	A	C	C	C	G	C	C	T	C	T	A	A
*V. chiangmaiensis* (MFLUCC 21-0065)	T	C	G	T	C	C	G	C	T	T	G	A	T	A	C	T	C	T	T
	216	221	229	232	233	235	239	421	432	442	446	447	448	451	461	465	557	558	
*V. bambusicola* (MFLUCC 11-0477)	T	A	A	C	T	T	G	T	C	T	C	T	C	C	T	G	T	T	
*V. chiangmaiensis* (MFLUCC 21-0065)	A	G	T	T	C	C	A	C	T	C	T	G	T	T	C	A	C	A	

**Table 5 jof-07-00891-t005:** Asexual morph comparison of the genera in *Clypeosphaeriaceae*, *Induratiaceae* and *Vamsapriyaceae*.

Family	Genus	Asexual Morph		
		Synnemata	Conidiogenous Cells	Conidia
*Vamsapriyaceae*	*Diabolocovidia*	Absent	Monoblastic, subcylindrical to clavate, pale brown	Catenate, acrogenous, brown, ellipsoid to obovoid, unseptate
	*Didymobotryum*	Present	Monotretic, integrated, terminal, clavate to cylindrical, pale brown to brown	Catenate, olivaceous brown to brown, cylindrical
	*Podosporium*	Present	Mono- or polytretic, subulate or cylindrical, darkly pigmented	Solitary, obclavate or bacilliform, multi-septate, brown to dark brown
	*Tretophragmia*	Present	Monotretic, subulate or cylindrical, darkly pigmented	Solitary, obclavate to fusiform or irregular in shape, multi-septate, dark brown
	*Vamsapriya*	Present	Monotretic, clavate to cylindrical	Catenate or solitary, acrogenous, cylindrical, oblong, fusiform or obclavate, brown to dark brown, septate
*Induratiaceae*	*Emarcea*	Absent	Integrated, terminal, pale brown, forming a rachis with numerous small, pimple-like denticles	Hyaline, smooth, falcate, granular, apex subobtuse, base truncate
	*Induratia*	Absent	Terminal, solitary or sometimes two celled at the ends of branches, cylindrical, pale brown, bearing inconspicuous denticles	Narrowly ellipsoidal to subglobose, hyaline

**Table 6 jof-07-00891-t006:** Sexual morph comparison of the genera in *Clypeosphaeriaceae*, *Induratiaceae* and *Vamsapriyaceae*.

Family	Genus	Sexual Morph	
		Asci	Ascospores
*Vamsapriyaceae*	*Vamsapriya*	8-spored, unitunicate, cylindrical, short pedicellate, with J+ apical ring	Apiosporous, fusiform to broad fusiform, hyaline, with sheath
*Induratiaceae*	*Emarcea*	8-spored, unitunicate, cylindrical, pedicellate, with J+ ring	Overlapping uniseriate, long fusiform, hyaline, 2-celled
	*Induratia*	8-spored, unitunicate, cylindrical, short pedicellate, with a J+ apical ring	Uniseriate, naviculate to ellipsoidal, mostly hyaline, constricted apiosporous
*Clypeosphaeriaceae*	*Aquasphaeria*	8-spored, unitunicate, cylindrical, with J- apical ring	Biseriate, cylindrical and ovoid, hyaline
	*Apioclypea*	8-spored, pedunculate, cylindrical, fissitunicate	Biseriate, fusiform, hyaline, with sheath
	*Brunneiapiospora*	8-spored, unitunicate, cylindrical, pedicellate with J+ or J− ascal ring	Hyaline to light brown apiospores with a mucilaginous sheath
	*Clypeosphaeria*	8-spored, unitunicate, cylindrical to broadly cylindrical, pedicellate, with J+ or J− ascal ring	Ellipsoidal to fusiform, unicellular to septate, hyaline to dark brown ascospores, sometimes with sheaths or appendages
	*Crassoascus*	8-spored, unitunicate, cylindrical, short pedicellate, with J+ ring	Bright brown to dark brown, multiseptate, fusiform ascospores, with hyaline refractive cap-like appendages at each end
	*Palmaria*	Cylindric to clavate, with a J− subapical ring	Apiosporous, hyaline, 1-septate, obclavate, with a mucilaginous sheath

## Data Availability

Not applicable.

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
