# Peer review of "Morphology and Phylogeny Reveal Vamsapriyaceae fam. nov. (Xylariales, Sordariomycetes) with Two Novel Vamsapriya Species"

_jof, 2021, doi:10.3390/jof7110891_

Round 1

Reviewer 1 Report

Dear editor and authors,

I congrats the authors for their effort to study. The manuscript is very important and defines an expected family to accommodate the genera that were as incertae sedis until now. The manuscript has relevance and quality enough to be published in the present form.

I have just two suggestions:

The Tub2 and RPB2 markers were used in the phylogenetic analyses. However, sequences from the new species were not provided here. That can be easy to provide.

Some other related family could be including in the phylogenetic tree (as Fasciatisporaceae) even knowing that this does not change the result about the new family proposed here.

So I wish all the best

Author Response

Dear Reviewer,

Thank you very much for your help to consummate our manuscript. We have finished the revision of this manuscript according to your suggestions.

Reviewer 2 Report

It was interesting to read the article “Morphology and phylogeny reveal Vamsapriyaceae fam. nov. (Xylariales, Sordariomycetes) with two novel Vamsapriya species”, with its excellent plates/illustrations. Here I would like to note some weak points that need attention, consideration and some necessary additions in order to support/improve the article. The remarks relate to the “Phylogenetic analyses” part, which calls for substantial improvement.

- Vamsapriyaceae is well supported in the combined dataset analysis, however a statement in the abstract that “Diabolocovidia, Didymobotryum and Vamsapriya formed a well-supported distinct clade which was basal to Clypeosphaeriaceae, Induratiaceae and Xylariaceae” is hard to understand/incorrect. The three families mentioned have a (very poorly supported; 50/-/0.98) sistergroup relationship to Vamsapriyaceae and none can be concluded to be ‘basal’ to the other (Figure 1).

- In addition to the four-region analysis it would be very desirable to present supplemental data showing separate single-region analyses to ensure that there are no important conflicts in topology.

- Overall the support for the new taxa treated is remarkably low, except for the family Vamsapriyaceae, which has a good support.

- It is surprisingly/worryingly low support for the data of Vamsapriya indica (77/89/-) and also for the genus Vamsapriya (-/61/-). Only three samples in the family Vamsapriyaceae have all four regions in the analyses, thus it would be very desirable/necessary to include an ITS-based phylogeny of Vamsapriya in the article.

Author Response

(The authors gave the same response as above.)

Round 2

Reviewer 2 Report

I would like to thank to the authors for their constructive response. The MS is well written and has excellent illustrations. I do believe in Vamsapriyaceae as a new family, based on the analyses and descriptions. My only concern is about how phylogenetic terminology has been used in the MS.

line 19: ‘and form an inter distinct clade in Xylariales’ – please, remove ‘inter’

line 113: ‘sequences are’

line 149: ITS is not a gene, so I would suggest using e.g. ‘markers’ or similar

line 153-154: ‘Xylariaceae, Induratiaceae and Clypeosphaeriaceae clustered together and formed a topmost clade,’ – please remove ‘and formed a topmost clade’

line 155: ‘Moreover, V. chiangmaiensis (MFLUCC 21-0065) formed a sister clade to V. yunnana’ - however, the support for this relationship in Fig. 1 is extremely poor and does not exist in Fig. 2

line 171: ‘indica taxa’ – please, remove ‘taxa’

Author Response

Dear reviewer,

Thanks for your valuable comments. 

We appreciate you taking your precious time in reviewing our paper and providing valuable comments. It was your valuable and insightful comments that led to possible improvements in the current version. The authors have carefully considered the comments and tried our best to address every one of them. We hope the manuscript after careful revisions meet your high standards comments.

Below we provide the point-by-point responses. All modifications in the manuscript have been used the“Track Changes”.

line 19: “and form an inter distinct clade in Xylariales” –removed “inter” and changed “an” to “a”

line 113: changed “sequences is” to “sequences are”.

line 149: Changed “The single gene trees (supplemental Figure 1,2,3,4) and the combined (LSU, rpb2, tub2 and ITS) gene tree (Figure 1) showed similar tree topology.” to “The single locus trees (supplemental Figure 1,2,3,4) and the multi-locus (LSU, rpb2, tub2 and ITS) tree (Figure 1) showed similar tree topology.”

line 153-154: “Xylariaceae, Induratiaceae and Clypeosphaeriaceae clustered together and formed a topmost clade,” –removed “and formed a topmost clade”

line 155: Changed “Moreover, V. chiangmaiensis (MFLUCC 21-0065) formed a sister clade to V. yunnana and V. uniseptata (GZCC 21-0892) is sister to V. indica.” to “Moreover, V. chiangmaiensis (MFLUCC 21-0065) formed a sister clade to V. yunnana, however, the support for this relationship in Figure. 1 is extremely poor and does not exist in Figure. 2, and V. uniseptata (GZCC 21-0892) is sister to V. indica.”

line 171: “indica taxa” –removed “taxa”

Thanks again.